# Usefulness of Brain Positron Emission Tomography with Different Tracers in the Evaluation of Patients with Idiopathic Normal Pressure Hydrocephalous

**DOI:** 10.3390/ijms21186523

**Published:** 2020-09-07

**Authors:** Maria Vittoria Mattoli, Giorgio Treglia, Maria Lucia Calcagni, Annunziato Mangiola, Carmelo Anile, Gianluca Trevisi

**Affiliations:** 1Department of Neurosciences, Imaging and Clinical Sciences, “G. d’Annunzio” Chieti-Pescara University, 66100 Chieti, Italy; mvittoriamattoli@yahoo.it (M.V.M.); annunziato.mangiola@ausl.pe.it (A.M.); 2Clinic of Nuclear Medicine, Imaging Institute of Southern Switzerland, Ente Ospedaliero Cantonale, 6500 Bellinzona, Switzerland; 3Academic Education, Research and Innovation Area, Ente Ospedaliero Cantonale, 6500 Bellinzona, Switzerland; 4Department of Nuclear Medicine and Molecular Imaging, Lausanne University Hospital, University of Lausanne, 1011 Lausanne, Switzerland; 5Dipartimento di Diagnostica per Immagini, Radioterapia Oncologica ed Ematologia, Fondazione Policlinico Universitario A. Gemelli IRCCS, Università Cattolica del Sacro Cuore, UOC di Medicina Nucleare, 00168 Rome, Italy; marialucia.calcagni@unicatt.it; 6Istituto di Medicina Nucleare, Università Cattolica del Sacro Cuore, 00168 Rome, Italy; 7Neurosurgery Unit, Santo Spirito Hospital, 65124 Pescara, Italy; trevisi.gianluca@gmail.com; 8Istituto di Neurochirurgia, Fondazione Policlinico Universitario A. Gemelli IRCCS, Università Cattolica del Sacro Cuore, 00168 Rome, Italy; carmelo.anile@policlinicogemelli.it

**Keywords:** hydrocephalous, ventriculo-peritoneal shunt, positron emission tomography, PET, biomarker, [^18^F]FDG, amyloid, perfusion, dopaminergic

## Abstract

Idiopathic normal pressure hydrocephalus (iNPH) is the only form of dementia that can be cured by surgery. Its diagnosis relies on clinical and radiological criteria. Identifying patients who can benefit from surgery is challenging, as other neurological diseases can be concomitant or mimic iNPH. We performed a systematic review on the role of positron emission tomography (PET) in iNPH. We retrieved 35 papers evaluating four main functional aspects with different PET radiotracers: (1) PET with amyloid tracers, revealing Alzheimer’s disease (AD) pathology in 20–57% of suspected iNPH patients, could be useful in predictions of surgical outcome. (2) PET with radiolabeled water as perfusion tracer showed a global decreased cerebral blood flow (CBF) and regional reduction of CBF in basal ganglia in iNPH; preoperative perfusion parameters could predict surgical outcome. (3) PET with 2-Deoxy-2-[^18^F]fluoroglucose ([^18^F]FDG ) showed a global reduction of glucose metabolism without a specific cortical pattern and a hypometabolism in basal ganglia; [^18^F]FDG PET may identify a coexisting neurodegenerative disease, helping in patient selection for surgery; postsurgery increase in glucose metabolism was associated with clinical improvement. (4) Dopaminergic PET imaging showed a postsynaptic D2 receptor reduction and striatal upregulation of D2 receptor after treatment, associated with clinical improvement. Overall, PET imaging could be a useful tool in iNPH diagnoses and treatment response.

## 1. Introduction

Normal pressure hydrocephalus (NPH) is a chronic condition among adults which is characterized by the clinical triad of gait disturbances, memory deficits and urinary incontinence, as well as radiological evidence of ventriculomegaly with disproportionally effaced superior frontal sulci and reduced callosal angle [1]. Despite the fact that it is known that NPH can be secondary to traumatic brain injuries, intracranial hemorrhages and infections, the majority of cases are idiopathic (iNPH), and their pathophysiology remaining puzzling. Since its recognition, NPH has been regarded as a form of dementia which is curable with surgery, namely through a ventriculo-peritoneal or ventriculo-atrial shunt. However, the selection of patients that will benefit from surgery remains difficult, with up to 40% of patients failing to respond [2].

Several adjuvant invasive tests have therefore been developed to predict shunt response, such as prolonged intracranial pressure (ICP) monitoring, cerebro-spinal fluid (CSF) subtraction tests (tap test, prolonged lumbar drainage), and infusion tests to study the CSF hydrodynamics (lumbar or ventricular Katzman test) [3,4,5,6,7]. Nonetheless, none of the above methods can fully predict shunt response. This may be due to the comorbidities burden of iNPH patients, including the concomitance of other neurodegenerative diseases [8]. Indeed, concomitant, biopsy proven hallmarks of AD have been reported in up to 68% of iNPH patients, and might negatively influence shunt-response [9]. Moreover, Parkinson-like, extrapyramidal motor disorders related to dopaminergic system dysfunction are also frequently reported in these patients [10,11]. Lastly, evidence has shown that iNPH is not simply the result of impaired CSF circulation, but that it has a complex pathogenesis involving brain vascular autoregulation deficits and decreased cerebral metabolism [12,13].

Positron emission tomography (PET) is a noninvasive functional imaging technique which is routinely used in clinical practice in the evaluation of patients affected by neurodegenerative diseases, including dementias and parkinsonian syndromes. With a portfolio of several radiotracers which are capable of assessing different cerebral metabolic functions or brain protein depositions, PET has been used in NPH patients to determine several cerebral functions: (1) the amyloid burden with different radiotracers i.e., 2-(4′-[^11^C]Methylaminophenyl)-6-hydroxybenzothiazole or [^11^C]Pittsburgh Compound B ([^11^C]PiB), 4-[(*E*)-2-[4-[2-[2-(2[^18^F]fluoranylethoxy)ethoxy]ethoxy]phenyl]ethenyl]-*N*-methylaniline ([^18^F]Florbetaben), and 2-[3-[^18^F]fluoranyl-4-(methylamino)phenyl]-1,3-benzothiazol-6-ol ([^18^F]Flutemetamol); (2) the cerebral perfusion with [^15^O]H_2_O; (3) the glucose metabolism with [^18^F]FDG; (4) and the integrity of the dopaminergic system with 3,5-dichloro-N-[[(2S)-1-ethylpyrrolidin-2-yl]methyl]-2-hydroxy-6-(^11^C)methoxybenzamide ([^11^C]Raclopride) and ^11^C-labelled 2β-carbomethoxy-3β-(4-fluorophenyl)tropane ([^11^C]CFT).

The aim of this systematic review is to assess the role of PET imaging with different radiotracers in iNPH patients according to published literature.

## 2. Results

A comprehensive literature search revealed 169 articles. Following the revision of both the title and the abstract, 133 article were excluded: 100 for not being within the field of interest of this review, 27 for being editorials or reviews, 6 for being case reports, and 1 for being a conference abstract reporting the same data as a separate, full-text manuscript. The full-text of the remaining articles was reviewed, and no additional study was included after screening the references. Finally, 35 articles comprising 674 patients were included in this systematic review (Figure 1).

The selected articles are categorized according to the protein deposition or cerebral function assessed by the different PET radiotracers utilized in the studies: (1) amyloid imaging; (2) perfusion imaging; (3) glucose metabolism imaging; or (4) dopaminergic imaging. The characteristics of all included articles are shown in Tables, according to the four main groups of studies.

### 2.1. Amyloid PET Imaging in iNPH

#### 2.1.1. Correlation between Radiotracer Uptake and Biopsy Samples in iNPH

At the beginning of the twenty-first century, amyloid PET imaging with [^11^C]PiB, and later with fluorinate radiopharmaceuticals (e.g., [^18^F]Florbetaben and [^18^F]Flutemetamol), represented a promising tool for noninvasive and in vivo evaluation of brain amyloid deposition in patients with suspected AD [14]. iNPH patients represented an ideal population to demonstrate the significant correlation between the amyloid PET uptake and AD-related lesion (Aβ aggregates) deposition upon histopathological examination. Indeed, iNPH patients have cognitive impairment and the AD is the most important type of dementia in differential diagnoses; furthermore, a notable percentage of patients with symptoms suggestive of NPH had histologically-proven AD-related pathological lesions [9]; finally, a cortical biopsy specimen can be made in these patients through the burr hole of ICP monitoring, a low-risk surgical procedure routinely performed in clinical practice that increases the diagnostic accuracy of iNPH.

Beside of a significant correlation between [^11^C]PiB PET findings and biopsy-proven AD-related lesions (Aβ aggregates), Leinonen et al. found 4/7 (57%) iNHP patients with concomitant Aβ-aggregates at biopsy and with [^11^C]PiB uptake at PET images [15]. Later, the same researchers performed a double-tracer PET imaging study with [^11^C]PiB and 1-[^18^F]fluoranyl-3-[2-[4-(methylamino)phenyl]quinolin-6-yl]oxypropan-2-ol (S-[^18^F]-THK-5117), a tau-specific ligand, aiming to investigate the relationship between PET and protein deposition (tau and amyloid) at biopsy in 14 iNPH patients. At biopsy, 50% patients had Aβ aggregates, 14% had both Aβ and p-tau lesions and 7% had p-tau lesions. Although a correlation between [^11^C]PiB PET and Aβ aggregates at biopsy was found, S-[^18^F]-THK-5117 did not show any statistically significant correlation with biopsy p-tau [16]. However, this tau tracer also binds to the monoamine oxidase-B (MAO-B) enzyme, which is known to be elevated in basal ganglia and could have led to off-target binding. This could constitute a limitation in the specificity of the study. Further studies with more specific Tau tracers should be conducted. A significant correlation between PET findings and Aβ aggregates at biopsy was also found using [^18^F]Flutemetamol. In particular, Wolk et al. found Aβ aggregates in 4/7 (57%) iNPH patients, resulting positive at [^18^F]Flutemetamol PET; a correlation between percentage of Aβ-aggregates and SUVratio (SUVvoi/SUVcerebellum) was also evident [17]. Similarly, Wong et al. found Aβ-aggregates at biopsy and positive [^18^F]Flutemetamol PET in 2/10 (20%) iNPH patients. The PET visual assessment had 100% in both specificity (Sp) and sensitivity (Se), and a good association between SUVratio and neuropathology was found [18]. Also Rinne et al. evaluated [^18^F]Flutemetamol PET in two series of iNPH patients undergoing biopsy during the shunting procedure. In these papers, they noted that Aβ-aggregates at biopsy were present in 24% and 29% of iNPH patients; the visual assessment had 75–100% in Se and 100% in Sp, and a significant correlation between SUVratio and Aβ-aggregates was found [19,20]. In the second study, the correlation between amyloid deposition and shunt response was also assessed, showing a variable response [20]. Finally, Leinonen et al. published a double-tracer study with [^18^F]Flutemetamol and [^11^C]PiB PET: 11 patients with final iNPH diagnosis underwent shunt, of whom two (18%) had Aβ-aggregates at biopsy; a significant association between the SUVratio of the two radiotracers was found (r = 0.97, *p* = 0.0003), and SUVratio of both radiotracers correlated with Aβ-aggregates at biopsy [21].

#### 2.1.2. Uptake in iNPH before Treatment

Once it had been established that amyloid PET imaging with [^11^C]PiB and [^18^F]Flutemetamol yielded similar results and effectively reflected the brain Aβ deposition at biopsy in iNPH patients, the researches’ focus moved to understanding the role of the concomitant AD-pathology, assessed by amyloid PET, in the pathophysiology of iNPH, comparing the PET findings with AD-pathology neurodegenerative disease or healthy control subjects (HC). To this end, Kondo et al. first compared the [^11^C]PiB uptake of 10 iNPH patients with those of seven AD patients. Three out of ten (30%) iNPH patients, all with mild cognitive impairment (MCI), showed [^11^C]PiB cortical uptake. The mean cortical SUVratio did not significantly differ between iNPH and AD patients, but a different [^11^C]PiB distribution was evident (high-convexity parasagittal areas in iNPH vs. frontal and parieto-temporal areas in AD) [22], that could be useful in the differential diagnoses. In 2017, Jiménez-Bonilla et al. visually analyzed the [^11^C]PiB uptake in 13 iNPH patients before shunt and, for comparison, in seven HC. PET scan was considered positive/equivocal in 5/13 (38%) iNPH patients, and a slight/mild cortical uptake was seen in all HC. Compared to HC, iNPH patients showed an increased cortical [^11^C]PiB retention and a lower white matter [^11^C]PiB retention, possible leading to lower contrast images [23]. Kang et al. compared the [^18^F]Florbetaben PET scan of 17 iNPH patients with 8 HC, showing significant increases in [^18^F]Florbetaben uptake in the bilateral frontal, parietal, and occipital cortices of iNPH patients compared to HC. In the iNPH group, right frontal [^18^F]Florbetaben uptake was found to be negatively correlated with right hippocampal volume, suggesting a relationship between frontal cortex [^18^F]Florbetaben uptake and hippocampal neuronal degeneration [24].

#### 2.1.3. Role in Prediction of Shunt Response in iNPH

Aiming to evaluate the clinical usefulness of amyloid PET imaging in the prediction of response to surgical shunt, Hirahoka et al. [25] performed amyloid PET imaging with 5-[(E)-2-[6-(2-fluoroethoxy)-1,3-benzoxazol-2-yl]ethenyl]-N-methyl-N-[^11^C]methyl-1,3-thiazol-2-amine ([^11^C]BF-227) in ten iNPH patients before shunt, and, for comparison, in ten AD patients and ten HC. According to the SUVratio values, iNPH patients with high-SUVratio (*n* = 5) and low-SUVratio (*n* = 5) did not significantly differ for baseline clinical assessments and clinical improvement after shunt, but a significant inverse correlation between SUVratios and cognitive improvements after shunt was observed. Jang et al. [26] evaluated the prognostic value of [^18^F]Florbetaben by analyzing the response to the tap test in 31 iNPH patients. Compared to patients with negative PET (*n* = 24), patients with positive PET (*n* = 7) showed worse improvement in gait scores after the tap test. Indeed, only 29% of [^18^F]Florbetaben positive patients were responders to the tap test. On the other hand, the tap test was positive in 83% of [^18^F]Florbetaben negative patients. [^18^F]Florbetaben PET was independently associated with the positive tap test response. Among the 14 patients undergoing shunting (responders at tap test and PET-negative), 12 (86%) showed improved symptoms. The authors concluded that [^18^F]Florbetaben PET scans can help determine which iNPH patients will benefit from shunt surgery by identifying concomitant AD. Rinne et al. [27] recently performed a retrospective [^11^C]PiB PET study in 21 suspected iNPH patients with ICP monitoring, aiming to compare the [^11^C]PiB uptake with Aβ and p-tau lesions at biopsy, the lumbar and ventricular CSF Aβ, the response to the shunt (performed in 15/21 patients), and the final clinical diagnosis of AD. Response to the shunt was seen in 13/15 patients, and AD was diagnosed in 8/21 patients during a median follow-up of 6 years. [^11^C]PiB uptake in the right frontal cortex (*p* = 0.60, *p* < 0.01) and the combined neocortical [^11^C]PiB uptake score (*p* = 0.61, *p* < 0.01) were associated with a higher Aβ load in the biopsy. Excluding one (1/15) outlier, [^11^C]PiB uptake was also associated with the ventricular CSF Aβ (*p* = −0.58, *p* = 0.03). The authors concluded that [^11^C]PiB PET can reliably detect simultaneous amyloid pathology among the iNPH patients; further studies will show whether amyloid PET can predict a clinical response to the shunt operation. In addition, the presence of Aβ pathology in patients with iNPH might also warrant treatment with current AD drugs.

Table 1 summarizes amyloid PET imaging findings.

### 2.2. Perfusion PET Imaging in iNPH

Despite the rapid evolution of magnetic resonance imaging (MRI) perfusion techniques and the wide availability of perfusion single photon emission computed tomography (SPECT), PET perfusion imaging with radiolabeled water is still considered the gold standard in the cerebral perfusion evaluation with imaging. Indeed, PET is the only technique which is intrinsically able to quantify, as absolute values (i.e., mL/min/gr), several perfusion parameters, such as cerebral oxygen utilization (rCMRO_2_), global and regional cerebral blood flow (CBF), blood volume (CBV), and oxygen extraction rate (OER).

#### 2.2.1. Perfusion Changes before Treatment in iNPH

In 2004, Owler et al. [28] measured the regional CBF with [^15^O]H_2_O PET coregistered with MRI in 17 NPH patients (12 idiopathic), and for comparison, in 12 HC. Mean CBF was significantly decreased in the cerebrum and cerebellum of iNPH patients compared to HC. Secondary NPH (posttraumatic or postinfective) also showed lower CBF in both cerebrum and cerebellum compared to HC, but these differences were not statistically significant. A regional analysis demonstrated that CBF was reduced in the basal ganglia and the thalamus, but not in white matter regions. In 2007, Miyamoto et al. [29] determined the CMRO_2_ and the CBF in patients with iNPH vs. HC at baseline in nine iNPH patients. CBF tended to decrease in frontal lobe and basal ganglia, whereas CMRO_2_ was increased in the frontal lobe and reduced in basal ganglia compared to 10 HC; regional CBV and OER showed no differences between iNPH and HC. The authors concluded that a reduction of oxygen metabolism in the basal ganglia might be one of the factors causing symptoms in iNPH; no particular pattern of oxygen metabolism in iNPH was evident in this study.

#### 2.2.2. Perfusion Changes during Infusion Test in iNPH

In 2004, Owler et al. [30] also showed how global and regional CBF (basal ganglia, thalamus, periventricular white matter) was further reduced at increased ICP values during a ventricular infusion test. Likewise, Momjian et al. [31] investigated the distribution of the regional peri- and para-ventricular white matter CBF using [^15^O]H_2_O PET coregistered with MRI in 12 iNPH patients at baseline (vs. 10 HC) and during the infusion test. At baseline, global CBF in iNPH was lower than in HC (28.4 vs. 33 mL/100 mL/min), with a progressive increase of white matter CBF at increasing distance from ventricles. The ventricular infusion test produced a reduction in cerebral perfusion pressure (CPP) and a significant decrease of the global CBF. The profile of the percentage changes in regional white matter CBF in iNPH patients showed a U-shaped relationship with distance from the ventricles, with a maximal decrease at a mean distance of 9 mm from the lateral ventricle wall, thus favoring corona radiata watershed territories strokes which are commonly seen in iNPH patients. These studies seem to confirm a defect in cerebral autoregulation underling the pathophysiological mechanism in iNPH during sleep-associated increased ICP [30,31].

#### 2.2.3. Perfusion Changes after Shunt Placement in iNPH

In 1986, Brooks and collaborators [32] measured the rCMRO_2_, rOER, rCBF, and the rCBV using PET with inhalation of tracer amounts of [^15^O]CO_2_, [^15^O]O_2_, and [^11^C]CO in 14 hydrocephalic patients (7 acute, 4 congenital, 3 iNPH) before and after surgical decompression, and in 20 HC. At baseline, all patients had a significantly reduced level of mean rCMRO_2_ and rCBF compared with HC, but acute hydrocephalus (AH) patients had elevated levels of rOER. After decompression, AH patients increased the rCBF, whereas chronic hydrocephalus (CH), including iNPH, showed no improvement in any perfusion parameter after shunt. Between 1998 and 2008, Klinge and collaborators published several studies aiming to investigate the cerebral perfusion measured with [^15^O]H_2_O PET before and after shunt in CH patients. The aim of their first study was to investigate whether the global and regional CBF was a reliable indicator for selecting patients for shunting, performing baseline and postshunt PET in 21 CH patients (18 iNPH). All the patients also underwent preoperative prolonged ICP monitoring and CSF infusion tests; only preoperative global CBF correlated with clinical outcome, as only patients with low global CBF clinically improved. ICP monitoring and resistance to CSF outflow did not correlate with outcome. Therefore, preoperative CBF may be helpful in predicting clinical outcome after shunting [33]. Subsequently, the same research group performed a similar study in 10 iNPH patients and, for comparison, in 10 HC. Besides the global CBF, the authors also investigated the cerebrovascular reserve (CVR) as the relative difference in the CBF between baseline and after vasodilatation with acetazolamide. At baseline, global CBF was significantly reduced in iNPH compared with HC, and CBF was significantly lower in shunt responders than in nonresponders; the baseline CVR was not significantly different between shunt responders and nonresponders. One week after shunting, the CVR significantly increased in responders and significantly decreased in nonresponders compared to respective baselines. At 7 months, both outcome groups showed an increase of CVR, with nonresponders returning to baseline values. A reduced baseline CBF before surgery did not indicate a poor prognosis, and baseline CBF and CVR were not predictive of clinical outcome. However, a decrease in the CVR soon after shunt placement was related to poor late clinical outcome [34]. Later, the same authors investigated the impact of cerebrovascular risk factors on perfusion parameters in 53 iNPH patients undergoing shunt placement (57% at low-risk). In high-risk patients, CVR at baseline was marginal in both outcome groups (~30%), but significantly increased in responders (64 vs. 31%) after shunt. In low-risk patients, CVR was lower in responders than in nonresponders (36 vs. 47%) at baseline, and deteriorated in nonresponders (29 vs. 47%) after shunt [35]. In 2002, the same research group also published three more studies that confirmed that at baseline, shunt responders had lower CBF but similar CVR to shunt nonresponders [33], being regional CBF and CVR reduced particularly in frontobasal cortex in responders [36]. After shunting, responders showed increased CVR [33], which was associated with both gait and cognitive improvement at 1 week and 7 months, respectively [37]. Lastly, in 2008, Kingle et al. [38] published a retrospective study questioning whether the functional status before and after shunt treatment might correlate with regional perfusion in 65 iNPH patients. At baseline, worse clinical status had significant correlation with a reduced tracer uptake in mesial frontal and anterior temporal areas. Comparison of mean global uptake before and after shunting did not reveal significant differences between shunt responders and nonresponders. In the mesial frontal areas, tracer uptake showed significant reciprocal changes in responders vs. nonresponders. Therefore, regional CBF alterations seem relevant to the NPH syndrome and to posttreatment functional changes. Also Miyamoto et al. [39] investigated the changes in CMRO_2_ with [^15^O]H_2_O PET before and after shunt; postshunt regional CMRO_2_ and OER was increased in good responders (*n* = 5), whereas the postshunt regional OER was reduced in poor responders (*n* = 3). The authors concluded that the improvement of regional CMRO_2_ correlated with the response to shunt, and that changes in regional OER might predict poor response to shunt.

Table 2 summarizes perfusion PET imaging findings.

### 2.3. Glucose Metabolism PET Imaging in iNPH

Normal pressure hydrocephalus is a potentially treatable form of dementia with an unpredictable outcome after shunt surgery. Since the middle of the 1980s, researchers have been evaluating cerebral glucose metabolism with [^18^F]FDG PET at baseline and after shunt, aiming to assess whether functional changes are able to predict response to shunt and the clinical outcome, thereby allowing them to better identify patients who would benefit from surgical procedures. Dynamic PET acquisition and quantitative assessment of metabolic rate of glucose (CMRglu, mL/min/100cc) was principally investigated, along with semiquantitative assessment.

#### 2.3.1. Metabolic Changes at Baseline in iNPH

The cerebral glucose metabolism evaluated with dynamic [^18^F]FDG PET was firstly investigated on 1985 by Jagust et al. [41], aiming to differentiate three NPH patients from 17 AD patients and seven HC. PET was performed only before treatment. All NPH patients improved after shunting. AD patients showed a typical temporoparietal hypometabolic pattern, while NPH showed a regular distribution of cortical uptake. However, a definite depression of CMRglu (mL/min/100cc) was found in all brain regions in NPH patients, and in temporoparietal cortex in AD. In 1995, Tedeschi et al. [42] measured rCMRglu in 18 iNPH submitted to shunt and frontal cortex biopsy. When compared with 11 HC, iNPH patients showed a significant rCMRglu reduction in all cortical and subcortical regions at baseline PET, with a large topographical heterogeneity (even after exclusion of six patients with AD-changes or cerebrovascular disease at biopsy). After shunt, only 33% patients improved (two with AD-changes and two with cerebrovascular changes in the biopsy sample), and the metabolic pattern did not differ between responders and nonresponders. These metabolic and histopathological heterogeneities may account for the high variability in the success rate of shunt surgery series.

In 2018, Townley et al. [43] evaluated semiquantitatively, with a voxel based statistical analysis, [^18^F]FDG PET scans in seven iNPH patients aiming to differentiate iNPH from 21 AD patients, 14 dementia with Lewy body/Parkinson disease dementia (DLB/PDD) patients, and seven behavioral fronto-temporal dementia (bvFTD) patients. When compared to HC and other pathologies, iNPH had significant regional hypometabolism in the dorsal striatum, involving the caudate and putamen bilaterally (even after partial volume correction), with preserved neocortical metabolism. This metabolic pattern may differentiate iNPH from degenerative diseases, and has the potential to serve as a biomarker for iNPH in future studies.

Finally, Miyazaki and collaborators [44] focused on the changes in regional cerebral glucose metabolism using [^18^F]FDG PET in a large population of prodromal iNPH patients. The authors distinguished among three groups of patients, according to the clinical and CT morphological findings, in order of NPH stages: (1) preclinical morphologic features of disproportionately enlarged subarachnoid-space hydrocephalus (PMD; *n* = 33); (2) asymptomatic ventriculomegaly with features of iNPH on MR (AVIM; *n* = 32); (3) iNPH (*n* = 12). Each group was compared with HC (*n* = 89). Measuring the median regional SUVratio in five regions (frontal lobes, temporal lobes, medial parietal lobes, striata, and thalami), the authors found that, compared to HC, the SUVratio was significantly lower in PMD, AVIM, and iNPH in the frontal and temporal lobes; it was significantly higher in PMD and AVIM in the medial parietal lobes; and it was significantly lower in the iNPH in the thalami and striata. Therefore, changes in brain glucose metabolism were mainly neocortical in the preclinical stage of iNPH, while metabolic decline in the basal ganglia was only detected in clinical iNPH.

#### 2.3.2. Metabolic Changes after Treatment in iNPH

In 1986, George et al. [45] studied five hydrocephalic patients (2 AH, 3 CH) with ^11^C-2-deoxyglucose or [^18^F]FDG PET before and after shunt. Despite a clinical improvement in all patients after shunting, the CMRglu increased in AH but decreased in CH. Calcagni and collaborators published two studies in 2011 and 2013 performing dynamic [^18^F]FDG PET in patients with iNPH before and after shunt [46,47]. In the first study, the authors evaluated the relationship between global CMRglu and clinical/neuropsychological assessment 3 days before and 1 week after shunt placement in 11 iNPH patients [46]. After shunting, the global CMRglu significantly increased (2.95 ± 0.44 vs. 4.38 ± 0.68, *p* = 10^−7^) in all iNPH patients (mean increase percentage of 48.7%), and all iNPH patients clinically improved with a significant correlation between the global CMRglu and clinical/neuropsychological assessment. The authors concluded that changes in the global CMRglu are promptly reversible after surgery, and that there is a relationship between the metabolic changes and clinical symptoms. In the second study, the authors performed a regional evaluation of CMRglu in 20 iNPH patients before and 1 week after shunt placement, and correlated the individual CMRglu variations with the clinical scale score assessment after shunting [47]. Baseline and postshunt regional CMRglu values were similar in right and left brain regions. After shunting, 17 patients (85%) improved both clinically (all scores decreased) and metabolically (CMRglu significantly increased in all regions), in contrast to the remaining patients that remained clinically stable but showed an increased CMRglu only in left frontal lobe, left putamen and right thalamus. However, no difference in global CMRglu at baseline was found between the two clinical outcome groups. Despite there was no correlation between clinical scales and pre- and post- operative CMRglu values, the study showed a significant inverse correlation between clinical assessment and the percentage of CMRglu variation. The authors concluded that iNPH is a global brain disease involving all cerebral regions in the same way, and that shunting also has a global effect on cerebral metabolism. Furthermore, individual variations of CMRglu were more important than absolute values, and were correlated with clinical status after shunting.

Table 3 summarizes glucose metabolism PET imaging findings.

### 2.4. Dopaminergic PET Imaging in iNPH

It is well known that gait disturbance is the most frequent among the triad of symptoms of iNPH [48]. However, differentiating iNPH from other neurologic diseases with hypokinetic type of gait disturbance, such as Parkinson’s disease (PD) and dementia with extrapyramidal symptoms, is sometimes a great challenge [49]. Only one research group has evaluated the dopaminergic function in iNPH with PET radiotracers. In the first study, Ouchi and collaborators [50] investigated the dopaminergic contribution to the pathophysiology of iNPH by performing a double-radiotracer study with [^11^C]CFT, a presynaptic marker binding to dopamine transporter, and [^11^C]Raclopride, a postsynaptic marker binding to D2 receptor, in eight iNPH patients and eight HC. They found a significant reduction in [^11^C]Raclopride binding in the putamen and nucleus accumbens and unchanged striatal [^11^C]CFT binding in iNPH. The dorsal putamen [^11^C]Raclopride binding correlated negatively with gait severity, whereas the nucleus accumbens [^11^C]Raclopride binding correlated positively with the emotional recognition score in the iNPH group. To summarize, iNPH shows preserved presynaptic activity in the nigrostriatal dopaminergic system, but a postsynaptic D2 receptor reduction. This postsynaptic D2 receptor hypoactivity in the dorsal putamen is associated with the severity of gait impairment in iNPH.

In 2007, the same study group investigated the plasticity of the striatum D2 receptor evaluating the difference of the [^11^C]Raclopride binding potential (BP) in eight iNPH patients before and 1 month after shunt [51]. After shunt, iNPH clinically improved, and the [^11^C]Raclopride BP increased in the nucleus accumbens and dorsal putamen; furthermore, the [^11^C]Raclopride BP increased in the striatum as a whole, correlating significantly with improved general cognitive ability. The authors concluded that striatal upregulation of D2 receptor after shunt is associated with amelioration of hypokinetic gait disturbance and anhedonic mentation in iNPH patients, indicating that the effect of VP shunting may reside in noninhibition of functionally suppressed D2 receptor in the striatum. 

Table 4 summarizes the dopaminergic PET imaging findings.

## 3. Materials and Methods

### 3.1. Search Strategy

A comprehensive computer literature search of the PubMed/MEDLINE, Cochrane library and Scopus databases was conducted in order to find relevant published articles on the role of PET and PET/CT in NPH. We used a search algorithm based on a combination of the following terms: (a) “Normal pressure hydrocephalus” or “NPH” and (b) “positron emission tomography” or “PET”. No beginning date limit was used; the search was updated until 30 May 2020. To expand our search, references of the retrieved articles were also screened for additional studies.

### 3.2. Study Selection

Studies investigating the role of PET and PET/CT in patients with iNPH were eligible for inclusion. Review articles or editorials, articles not in the field of interest of this review, case reports and preclinical studies were excluded. Only studies including PET or PET/CT performed in at least 3 iNPH patients were included. Two researchers (M.V.M. and G.T.) independently reviewed the titles and abstracts of the retrieved articles, applying the above inclusion and exclusion criteria.

### 3.3. Data Abstraction

For each included study, we collected: the basic details of study (authors and year of publication); patient characteristics (number, gender and mean age); technical aspects of PET imaging (kind of radiopharmaceutical, kind of analysis, parameters evaluated, and PET timing); and neurosurgical aspects (kind of hydrocephalous and surgical procedure performed). Finally, the primary endpoint and the main findings of all articles included in this review are shown in the Results section.

Studies reporting data not only on iNPH but also from other forms of hydrocephalus (e.g., secondary NPH, longstanding overt hydrocephalus, obstructive NPH, etc.) were not excluded, but only results about iNPH patients were included in the present study.

## 4. Summary

To summarize the information provided in the retrieved articles concerning PET imaging in iNPH patients:PET with amyloid tracers, revealing AD in 20–57% of suspected iNPH patients, could be useful in prediction of surgical outcome;PET with perfusion tracers showed a global decreased CBF and regional reduction of CBF in basal ganglia in iNPH patients; preoperative perfusion parameters could predict surgical outcome;PET with [^18^F]FDG showed a global reduction of glucose metabolism without a specific cortical pattern and a hypometabolism in basal ganglia in iNPH patients; [^18^F]FDG PET may identify a coexisting neurodegenerative disease, helping in the patient selection for surgery; postsurgery increase in glucose metabolism was associated with clinical improvement.Dopaminergic PET imaging showed a postsynaptic D2 receptor reduction and a striatal upregulation of D2 receptors after treatment, associated with clinical improvement.

## 5. Limitations

The main limitations of our study are related to the heterogeneity of the retrieved studies, in particular, with respect to number of patients, PET evaluation techniques and timing of the imaging. Indeed, the heterogeneity of the timing of perfusion or glucose metabolism PET, performed either before or after the surgical procedure, could affect the prognostic value of PET imaging. However, the aim of these studies was different, as preoperative only PET studies aimed to detect the static picture of the disease and possible markers of response to shunt, while pre- and post- op PET studies aimed to increase knowledge on treatment-induced changes and to detect dynamic changes which could predict longer-term outcome.

Moreover, prevalence of beta amyloid deposit burden could be limited by heterogeneous brain tissue sampling (frontal/parietal) and by the reduced amount of tissue sampled in the setting of ventricular catheter placement. However, the aim of some included papers was to validate beta amyloid tracers using biopsy samples as a reference standard, while the aim of others was to assess whether the presence/absence of these deposits was associated with outcome after shunting.

Finally, a common limitation of the retrieved studies was the poor focus on possible differences in neuropsychological assessments of patients included in the study. This limits the possibility to evaluate whether a basal difference in cognitive impairment exists between iNPH patients with and without concomitant beta amyloid deposits, and to fully assess the outcome after treatment in these patients.

## 6. Future Perspectives

Currently, the diagnostic and prognostic value of PET imaging with different tracers in iNPH patients seems promising, but these methods are not part of the routine selection procedures for shunt surgery in iNPH. Furthermore, current guidelines for the diagnosis of iNPH and 2020 ACR appropriateness criteria do not recommend PET imaging for iNPH evaluation [52,53]. Nevertheless, as reported in our review, recent studies have shown a promising role of PET with [^18^F]FDG and amyloid tracers for the evaluation of iNPH patients [54]. These findings should be confirmed by more large-scale prospective studies to support the role of these imaging methods in the management of iNPH patients. Moreover, studies with a combination of beta amyloid and glucose metabolism imaging could be useful in evaluations of concomitant AD in iNPH and to improve assessments of expected outcomes of surgical treatment.

## 7. Conclusions

Positron emission tomography is a molecular imaging technique which is able to provide functional information regarding perfusion, glucose metabolism and amyloid deposition in patients with iNPH. The value of this technique is its unique, intrinsic ability to provide absolute quantification of functional parameters, such as perfusion or metabolism, measured in mL/min/cc. PET radiotracers can show metabolic and functional changes that occur in spite of the morphological ones, giving earlier and prodromal information. PET with different tracers seems to be useful to understand the pathophysiology of iNPH in order to improve the selection of patients for surgery, determining the optimal timing for therapeutic intervention, and potentially making it possible to predict the expected outcome, both in terms of global iNPH symptoms and in specific iNPH domains such as cognitive impairment and gait deficits. Nevertheless, more studies are needed to validate these findings.

## Figures and Tables

**Figure 1 ijms-21-06523-f001:**
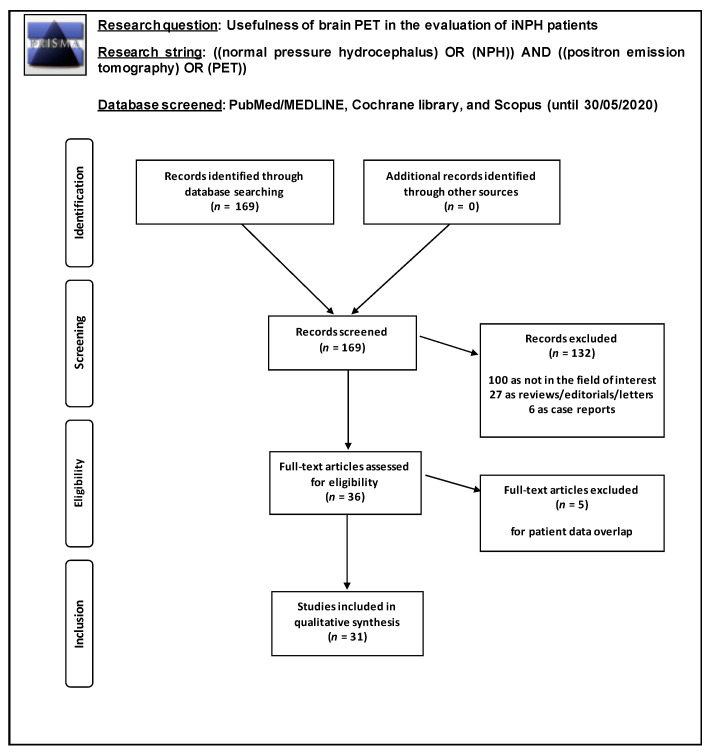
Flow chart of the search for the systemic review on the usefulness of brain positron emission tomography (PET) with different tracers in patients with iNPH patients.

**Table 1 ijms-21-06523-t001:** Amyloid PET imaging in NPH patients.

Authors	N. of Pts	Tracer	Analysis of PET Images	PET Timing	Surgical Procedure	Primary Endpoint	Main Findings
Leinonen et al., (2008) [15]	10	[^11^C]PiB	Semiquantitative (SPM)	Postshunt	Shunt + frontal biopsy	To compare [^11^C]PiB findings in patients with and without AD lesions in frontal cortical biopsy.	7/10 pts had diagnosis of iNPH; 3/7 iNPH had Aβ aggregates and PET positive (higher uptake in the lateral frontal and lateral temporal cortices, anterior and posterior cingulate gyri, and caudate nucleus).
Wolk et al., (2011) [17]	7	[^18^F]Flutemetamol	Qualitative; semiquantitative (SUVratio)	Postshunt	Shunt + frontal biopsy	To compare tracer uptake with amyloid level at biopsy.	4/7 pts had Aβ aggregates and PET positive. A correlation between Aβ deposition and SUVratio in the VOI contralateral to the biopsy site was evident.
Rinne et al., (2012) [19]	52	[^18^F]Flutemetamol	Qualitative and semiquantitative (SUVratio)	Before and postshunt	Shunt + frontal biopsy	To determine the association between tracer uptake and neuritic plaques and fibrillar amyloid β.	49/52 had sufficient data for analyses. 14/49 pts had Aβ deposition. Significant association between SUVratio and neuritic plaque burden. Se 93% and Sp 100%. [^18^F]Flutemetamol offers a noninvasive tool that may be useful for identifying the presence of AD-related lesions.
Wong et al., (2013) [18]	12	[^18^F]Flutemetamol	Qualitative; semiquantitative (SUVratio)	Baseline	Shunt + parietal biopsy	To determine the association between [^18^F]Flutemetamol uptake and amyloid levels at biopsy.	10/12 pts had biopsy sample; 2/10 had Aβ aggregates. Qualitative PET had 100% in Se and Sp. The biopsy site SUVratio was correlated with the biopsy Aβ level and with the composite SUVratio. Amyloid burden in the biopsied region was representative for the entire cortex.
Leinonen et al., (2013) [21]	15	[^18^F]Flutemetamol ([^11^C]PiB in 7 pts)	Qualitative; semiquantitative (SUVratio)	Postshunt	Shunt + frontal biopsy	To determine the correlation between [^18^F]Flutemetamol and Aβ aggregates levels.	11/15 pts had diagnosis of iNPH; 2/11 had Aβ aggregates. [^18^F]Flutemetamol and [^11^C]PiB SUVratios correlated with Aβ aggregates. NPH pts with PET positive may warrant treatment with AD drugs;
Kondo et al., (2013) [22]	10 (+ 7 AD)	[^11^C]PiB	Semiquantitative (SUVratio)	Baseline	None	To elucidate the distribution of [^11^C]PiB in iNPH and clarify the differences with AD pts.	3/10 iNPH pts had ↑ [^11^C]PiB uptake (all with MCI). iNPH vs. AD: similar mean cortical SUVratio but different [^11^C]PiB distribution (high-convexity parasagittal areas in iNPH vs. frontal and parieto-temporal areas in AD). [^11^C]PiB could be useful to differentiate iNPH from AD.
Rinne et al., (2013) [20]	17	[^18^F]Flutemetamol	Qualitative; semiquantitative (SUVratio)	Baseline	Shunt + frontal biopsy	To determine the association between [^18^F]Flutemetamol uptake and Aβ aggregates levels.	4/17 pts had Aβ aggregates and [^18^F]Flutemetamol uptake; Aβ aggregates correlated with SUVratios. Qualitative PET showed 75–100% Se and 100% Sp. In PET positive pts, the response to shunt was variable (good in 1, fair in 1 and transient in 2 pts).
Hiraoka et al., (2015) [25]	10 (+ 10 HC + 10 AD)	[^11^C]BF-227	Semiquantitative (SUVratio)	Baseline	Shunt	To evaluate amyloid deposition preshunt and its association with postshunt response.	5/10 pts had high SUVratios (similar to AD); 5/10 pts had low SUVratios (similar to HC). A significant inverse correlation between neocortical SUVratios and cognitive improvements after shunt surgery was observed.
Jiménez-Bonilla et al., (2017) [23]	13 (+ 7 HC)	[^11^C]PiB	Qualitative	Baseline	None	To compare the [^11^C]PiB uptake pattern in iNPH pts and in HC.	8/13 negative, 3/13 positive, 2/13 equivocal PET scans. 5/13 iNPH pts showed [^11^C]PiB uptake higher than HC. 4/7 HC had diffuse slight uptake, 3/7 HC had mild uptake in at least one region. WM uptake of iNPH scored lower than in HC.
Jang et al., (2018) [26]	31	[^18^F]Florbetaben	Qualitative; semiquantitative (SUVratio)	Baseline	tap test and shunt	To evaluate the prognostic value of [^18^F]Florbetaben by analyzing the response to the tap test.	7/31 pts had PET positive. PET negative pts (24/31) had ↑ % of R at tap test and showed ↑ gait scores. PET positivity and CSF p-tau were independently associated with the response to tap test. Amyloid PET can help to determine which iNPH pts will benefit from shunt by discriminating concomitant AD.
Kang et al., (2018) [24]	17 (+ 8 HC)	[^18^F]Florbetaben	Semiquantitative (SUVratio)	Baseline	None	To investigate the cortical uptake in iNPH and HC; the relationships between [^18^F]Florbetaben uptake, hippocampal volume and clinical symptoms.	iNPH pts showed ↑ uptake in bilateral frontal, parietal, and occipital regions. In iNPH, right hippocampal volume was negatively correlated with right frontal uptake. ↑ uptake significantly correlated with ↑ CDRS score in the right occipital cortex. iNPH might exhibit a characteristic pattern of cortical uptake.
Leinonen et al., (2018) [16]	14	S-[^18^F]-THK-5117 (tau deposit) and [^11^C]PiB	Semiquantitative (Voxel-based statistical analysis)	Postshunt	Shunt + frontal biopsy	To evaluate the association between CSF, S-[^18^F]THK-5117, and [^11^C]PiB PET against tau and amyloid lesions.	7/14 pts had Aβ lesions, 2/14 pts had both Aβ and tau lesions, 1/14 had tau lesions. Pts with Aβ lesions had higher [^11^C]PiB uptake. ^18^F-THK-5117 uptake did not correlate with biopsy tau level or CSF p-tau or t-tau.
Rinne et al., (2019) [27]	21	[^11^C]PiB	Semiquantitative (time-activity curves ratio)	Baseline	Shunt + frontal biopsy	To compare the [^11^C]PiB uptake to Aβ and tau lesions and CSF Aβ.	11/21 pts had Aβ aggregates; 15 pts underwent shunt (13 clinically improved). AD in 8/21 pts. [^11^C]PiB uptake was associated with a higher Aβ aggregates and CSF Aβ.

Legend: AD = Alzheimer’s disease; CDRS = clinical dementia rating scale; CSF = cerebrospinal fluid; HC = healthy controls; iNPH = idiopathic normal pressure hydrocephalus; MMSE = mini mental scale examination; NR = nonresponder; pts = patients; R = responder; Se = sensitivity; Sp = specificity; SPM = statistical parametric mapping; SUV = Standardized uptake value; VOI = volume of interest; WM = white matter; ↑ = elevated.

**Table 2 ijms-21-06523-t002:** Perfusion PET imaging in NPH patients.

Authors	N. of Pts	Tracer	Analysis of PET Images	PET Timing	Surgical Procedure	Primary Endpoint	Main Findings
Brooks et al., (1986) [32]	14 (+ 20 HC)	[^15^O]CO_2_ [^15^O]O_2_ [^11^C]CO	Quantitative (regional CMRO_2_, OER, CBF, CBV)	Baseline and postsurgical decompression	Surgical decompression	To evaluate CMRO_2_, CBF, and OER in hydrocephalous pts (obstructive and idiopathic) before (vs. HC) and after surgical decompression.	All hydrocephalous pts had ↓ CMRO_2_and CBF (than HC); iNPH pts had CBF/CMRO_2_ matched and normal OEF at baseline, and showed no improvement in CBF or CMRO_2_ or cognitive function after surgical intervention.
Klinge et al., (1998) [40]	21	[^15^O]H_2_O	Quantitative (global and regional CBF)	Baseline and postshunt	Shunt	To evaluate whether CBF was an indicator in selecting pts to undergo shunt.	After shunt, 12/21 pts clinically improved. Pts with lower global CBF showed clinical improvement after 7 months (in contrast with pts with higher CBF); baseline global CBF discriminated in terms of clinical outcome; CBF may be helpful in evaluating the utility of shunt.
Klinge et al., (1999) [34]	10 (+ 10 HC)	[^15^O]H_2_O	Quantitative (global CBF and CVR)	Baseline and postshunt	Shunt	To evaluate CBF and CVR in chronic hydrocephalus (vs. HC) after shunt.	Baseline global CBF was ↓ compared to HC; after shunt, 5/10 pts clinically improved; baseline CBF and CVR were not predictive of clinical outcome; early improvement in CVR after shunt indicated a good prognosis.
Klinge et al., (2002) [35]	53	[^15^O]H_2_O	Quantitative (global CBF and CVR)	Baseline and postshunt	Shunt	To investigate the impact of cerebrovascular risk factors in cerebral hemodynamics.	Pts stratified into HR and LR groups; in HR, baseline CBF was ↓ in R, while CVR was marginal in both R and NR; after 1 week, in HR, CVR of R improved; in LR, CVR of NR decreased.
Klinge et al., (2002) [33]	60	[^15^O]H_2_O	Quantitative (global CBF + CVR)	Baseline and postshunt (1 week and 7 months)	Shunt	To evaluate CBF and CVR in chronic hydrocephalus after shunt	After shunt, 31/60 pts clinically improved (R); at baseline, R had ↓ CBF than NR, and CVR was not different between R and NR; after shunting, CVR decreased in NR and increases in R; CBF might substantially contribute to selection of shunt candidates, and neurological sequels may be related to early regeneration of the hemodynamic reserve.
Klinge et al., (2002) [36]	11	[^15^O]H_2_O	Quantitative + SPM (regional CBF + CVR)	Baseline	Shunt	To find out if regional CBF and CVR may indicate shunt response in idiopathic chronic hydrocephalus	CBF ↓ in the frontobasal cortex in R than in NR; CVR ↓ in a variety of cortical regions in R compared with NR, including frontobasal cortex (most ↓ in temporo-dorsal and limbic cortical regions); baseline hemodynamics displayed a regional profile of reduced CBF and CVR in pts with shunt response.
Klinge et al., (2002) [37]	27	[^15^O]H_2_O	Quantitative (global and regional CBF + CVR)	Baseline and postshunt (1 week and 7 months)	Shunt	To evaluate the relationship of neuropsychological deficits before and after shunting with dynamics in cerebral blood flow.	No relationship of test profiles with baseline CBF or CVR; after 1w, improvement of gait related to ↑ CVR; after 7m, improvement in mental tests related to ↑ CVR; neurological sequels after shunting may depend on consecutive improvement of hemodynamics
Owler et al., (2004) [28]	17 (+ 12 HC)	[^15^O]H_2_O	Quantitative (global and regional CBF)	Baseline	None	To study the regional CBF in NPH with PET coregistered with MRI (vs. HC).	CBF was significantly ↓ in the cerebrum and cerebellum of iNPH pts than HC; CBF ↓ in basal ganglia and thalamus (not in WM regions); the role of the basal ganglia and thalamus in iNPH may be more prominent than currently appreciated.
Momjian et al., (2004) [31]	12 (+ 10 HC)	[^15^O]H_2_O	Quantitative (regional CBF)	Baseline and during infusion test (plateau of raised ICP)	Infusion test	To investigated the distribution of peri- and paraventricular WM CBF in NPH at baseline (vs. HC) and during the infusion test.	At baseline, global CBF in iNPH was ↓ than in HC; in iNPH, the profile of the regional WM CBF at baseline showed an ↑ with distance from the ventricles. In 10 pts, infusion test caused ↓ in CPP and in global CBF; WM CBF was ↓ in iNPH, with an abnormal gradient from the lateral ventricles to the subcortical WM.
Owler et al., (2004) [30]	15	[^15^O]H_2_O	Quantitative (global and regional CBF)	Baseline and during infusion test (plateau of raised ICP)	Infusion test	To investigate the global and regional changes in CBF with changes in CSF pressure (infusion test)	With ↑ in CSF pressure, global CBF was ↓ (including cerebellum). rCBF ↓ in thalamus, basal ganglia, and in WM regions (correlated with changes in the CSF pressure and with proximity to the ventricles).
Miyamoto et al., (2007) [39]	8	[^15^O]H_2_O	Quantitative (regional CMRO_2_ and OEF)	Baseline and postshunt (3 months)	Shunt	To investigate the changes in cerebral oxygen metabolism before and after shunt.	After shunt, 5/8 good R and 3/8 poor R; the postshunt regional CMRO_2_ is ↑ in good R; the postshunt regional OEF is ↓ in the poor R. The improvement of r CMRO_2_ correlated with shunt response; changes in rOEF might predict poor shunt response.
Miyamoto et al., (2007) [29]	9 (+ 10 HC)	[15O]H2O	Quantitative (regional CBF, CBV, OEF and CMRO_2_)	Baseline	Shunt	To determine the cerebral oxygen metabolism and the cerebral blood flow in patients with iNPH vs. HC.	↓ CBF in frontal lobe and basal ganglia; ↑ CMRO_2_ in frontal lobe compare to HC (but CMRO2 in basal ganglia of iNPH was ↓); rCBV and rOEF showed no differences. Reduction of O_2_ metabolism in basal ganglia might cause symptoms in iNPH; no particular pattern of oxygen metabolism in iNPH.
Klinge et al., (2008) [38]	65	[^15^O]H_2_O	Quantitative + SPM (global and regional CBF, CVR)	Baseline and postshunt (7–10 days)	Shunt	To evaluate whether the functional status before and after shunt treatment might correlate with local blood flow in iNPH pts.	At baseline, ↑ clinical score correlated with ↓ tracer uptake in mesial frontal and anterior temporal areas; at postshunt, in the mesial frontal areas ↑ uptake in R and ↓ uptake in NR. Regional blood flow changes are relevant to NPH and to postshunt functional changes.

Legend: CBF = cerebral blood flow; CBV = cerebral blood volume; CMRO_2_ = cortical oxygen utilization; CPP = cerebral perfusion pressure; CSF = cerebrospinal fluid; CVD = cerebrovascular disease; CVR = cerebrovascular reserve after administration of 1 gr of Acetazolamide; HC = healthy controls; HR = high-risk; ICP = intracranial pressure; LR = low-risk; NPH = normal pressure hydrocephalus; NR = non responder; O2 = oxygen; pts = patients; R = responder; rOER = oxygen extraction rate; SPM = statistical parametric mapping; WM = white matter; ↓ = reduced; ↑ = elevated.

**Table 3 ijms-21-06523-t003:** Glucose metabolism PET imaging in NPH patients.

Authors	N. of Pts	Tracer	Analysis of PET Images	PET Timing	Surgical Procedure	Primary Endpoint	Main Findings
Jagust et al., (1985) [41]	3 (+ 17 AD + 7 HC)	[^18^F]FDG	Quantitative (rCMRglu)	Baseline	Shunt	To differentiate patients with NPH from AD	Both AD and NPH showed lower CMRglu than HC. The hypometabolic pattern was different: AD had bilateral temporo-parietal hypometabolism, NPH had global hypometabolism.
George et al., (1986) [45]	5	[^18^F]FDG	Quantitative (gCMRglu)	Baseline and postshunt	Shunt	To study the glucose metabolism before and after shunt	After shunt, all patients clinically improved. After shunt, ↑ CMRglu in acute hydrocephalus pts; ↓ in chronic hydrocephalus pts despite clinical improvement.
Tedeschi et al., (1995) [42]	18 (+ 11 HC)	[^18^F]FDG	Quantitative (rCMRglu) and semiquantitative	Baseline	Shunt + frontal biopsy	To evaluate CMRglu at baseline in iNPH underwent shunt (vs. HC)	↓ rCMRglu in all regions in NPH than HC; no typical metabolic pattern of NPH; AD/CVD in 6/18 pts; after shunt, 6/18 pts clinically improved (2 AD and 2 CVD); no significant different in metabolic pattern between R (6/18) and NR (12/18); high metabolic and histopathological heterogeneity.
Calcagni et al., (2012) [46]	11	[^18^F]FDG	Quantitative (gCMRglu)	Baseline and postshunt	Shunt	To evaluate the gCMRglu and clinical assessment before and after shunt	After shunt, ↑ gCMRglu in all pts, and ↓ in all clinical scale scores (significant correlation between CMRglu and clinical assessment).
Calcagni et al., (2013) [47]	20	[^18^F]FDG	Quantitative (rCMRglu; gCMRglu)	Baseline and postshunt	Shunt	To evaluate the rCMRglu before and after shunt and to correlated the CMRglu with the clinical scale scores.	No differences in CMRglu between right and left brain regions at baseline and postshunt; at postshunt, R (17/20) had ↓ scale scores and ↑ CMRglu in all regions; in NR (3/20) stable scale scores and ↑ CMRglu in only 3 regions (left frontal, left putamen, and right thalamus). At baseline, no difference in gCMRglu between R and NR. At postshunt, individual % variations of CMRglu correlated with clinical scores.
Townley et al., (2018) [43]	7 (+ 21 HC + 14 AD + 14 DLB/PDD + 7 bvFTD)	[^18^F]FDG	Semiquantitative + SPM	Baseline	None	To differentiate iNPH from neurodegenerative disorders.	iNPH group had significant hypometabolism in the dorsal striatum (tan other groups), involving the caudate and putamen bilaterally; no specific pattern of significant cortical hypometabolism. This pattern may differentiate iNPH from degenerative diseases.
Miyazaki et al., (2019) [44]	12 (+ 33 PMD + 32 AVIM + 89 HC)	[^18^F]FDG	Semiquantitative (SUVratio)	Baseline	None	To evaluate the changes in regional cerebral glucose metabolism with respect to the characteristic morphologic features of iNPH.	In the frontal and temporal lobes, SUVr in PMD, AVIM, and iNPH groups were significantly lower than HC; in parietal lobes, the SUVr were significantly higher in PMD and AVIM groups; in the thalami and striata, the SUVr were significantly lower in iNPH. Changes in glucose metabolism in the cortices in preclinical iNPH; basal ganglia hypometabolism only in clinical iNPH.

Legend: AD = Alzheimer’s disease; AVIM = asymptomatic ventriculomegaly with features of iNPH on magnetic resonance; bvFTD = behavioral fronto-temporal dementia; CVD = cerebrovascular disease; DLB/PDD = dementia with Lewy body/Parkinson disease dementia; gCMRglu = global cerebral metabolic rate for glucose; HC = healthy controls; iNPH = idiopathic normal pressure hydrocephalus; NR = nonresponder patients; PMD = preclinical morphologic features of disproportionately enlarged subarachnoid-space hydrocephalus; pts = patients; R = responder patients; rCMRglu = regional cerebral metabolic rate for glucose; SPM = statistical parametric mapping; SUVratio = standardized uptake value ratio; ↓ = reduced; ↑ = elevated.

**Table 4 ijms-21-06523-t004:** Dopaminergic PET imaging in NPH patients.

Authors	N. of Pts	Tracer	Analysis of PET Images	PET Timing	Surgical Procedure	Primary Endpoint	Main Finding s
Ouchi et al., (2007) [50]	8 (+ 8 HC)	[^11^C]CFT (presynaptic marker)	Quantitative (BPs)	Baseline	None	To differentiate impaired gait seen in iNPH from parkinsonian gait; to investigate dopaminergic contribution to iNPH pathophysiology (vs. HC)	↓ [^11^C]Raclopride BP in putamen and nucleus accumbens than HC; equal [^11^C]CFT BP in iNPH and HC; putamen [^11^C]Raclopride BP negatively correlated with gait severity; nucleus accumbens [^11^C]Raclopride BP positively correlated with emotional recognition score; postsynaptic receptor reduction and preserved presynaptic status found in NPH.
		[^11^C]Raclopride (postsynaptic marker)					
Nakayama et al., (2007) [51]	8	[^11^C]Raclopride (postsynaptic marker)	Quantitative (BPs)	Baseline and postshunt (1 month)	Shunt	To evaluate the plasticity of D2 receptor in treating iNPH pts with ventriculoperitoneal shunt	All pts clinically improved after shunt. At postshunt PET, ↑ in [^11^C]Raclopride BP in nucleus accumbens and dorsal putamen; ↑ [^11^C]Raclopride BP in striaum correlated with improvement in general cognitive ability.

Legend: BP = binding potential; pts = patients; HC = healthy controls; ↓ = reduced, ↑ = elevated.

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
