# Peer review of "Usefulness of Brain Positron Emission Tomography with Different Tracers in the Evaluation of Patients with Idiopathic Normal Pressure Hydrocephalous"

_ijms, 2020, doi:10.3390/ijms21186523_

Round 1

Reviewer 1 Report

Comments (IJMS-895145)

“Usefulness of Brain Positron Emission Tomography with Different Tracers in the Evaluation of Patients with Idiopathic Normal Pressure Hydrocephalous” by Maria Vittoria Mattoli, Giorgio Treglia, Maria Lucia Calcagni, Annunziato Mangiola, Carmelo Anile and Gianluca Trevisi.

The authors have reviewed 35 articles which evaluate four main functional aspects with different PET radiotracers; amyloid tracers, radiolabeled water as perfusion tracer, [18F]FDG, and dopaminergic PET tracers (D2 and dopamine transporter tracers). This kind of review might help the clinicians to plan possible PET studies for iNPH patiens and assess the usefulness of the studies.

Comments:

  • tracer names: The nomenclature of presented radiopharmaceuticals should follow the guidelines presented in:

“Consensus nomenclature rules for radiopharmaceutical chemistry — Setting the record straight, Heinz H.Coenen, Antony D.Gee, MichaelAdam, GunnarAntoni, Cathy S.Cutler, YasuhisaFujibayashi, Jae MinJeong, Robert H.Mach, Thomas L.Mindt, Victor W.Pike, Albert D.Windhorst, published in Nucl Med Biol, 2017, https://doi.org/10.1016/j.nucmedbio.2017.09.004”. It is strongly recommend that all manuscripts meet these guidelines upon submission.

This means that the tracer names should be written: [18F]FDG, [18F]Florbetaben, [18F]Flutemetamol, [11C]PIB, [11C]raclopride, [15O]H2O etc.. i.e. the used radionuclide in square brackets and 18, 11, etc. as superscripts. But without square brackets if e.g. 18F-labelled.

  • The full chemical name has been added to [18F]FDG and [11C]CFT but not to the other tracers. Please check this inconsistency. Please use the IUPAC-name if possible; e.g. 2-Deoxy-2-[18F]fluoroglucose for [18F]FDG.
  • Lines 72 and 73, 204 and 205, 393, 412. Please correct the name for 11C-labelled 2β-carbomethoxy-3β-(4-fluorophenyl)tropane. Instead of the letter b, the Greek letter beta should be used. In the text the name should be mentioned only once and thereafter to use the abbreviation [11C]CFT.
  • The authors mention also a tau-tracer, S-[18F]THK-5117. This could perhaps be addressed separately. It may also be worth mentioning that this tracer has subsequently been shown to bind to MAO-B as well. In other words, the study should be repeated with some other, more specific tau tracer.
  • Keywords: Please check if the keywords are the best ones.
  • Line 138. Misspelled Rinne et al.
  • PiB has been written in two different ways. PiB is more common than Pib. Please check
  • Line 243 and table 2, please check if there should be [15O]CO instead of [11C]CO. (Ref. 32 Brooks et al.)
  • Please check that all reference titles are written exactly like in the original paper, e.g. that the superscripts and square brackets are correct. Please, also use the method described in the IJMS to abbreviate the journal names.

Author Response

Reply to the Reviewers’ comments:

- tracer names: The nomenclature of presented radiopharmaceuticals should follow the guidelines presented in: “Consensus nomenclature rules for radiopharmaceutical chemistry — Setting the record straight, Heinz H.Coenen, Antony D.Gee, MichaelAdam, GunnarAntoni, Cathy S.Cutler, YasuhisaFujibayashi, Jae MinJeong, Robert H.Mach, Thomas L.Mindt, Victor W.Pike, Albert D.Windhorst, published in Nucl Med Biol, 2017”. It is strongly recommend that all manuscripts meet these guidelines upon submission. This means that the tracer names should be written: [18F]FDG, [18F]Florbetaben, [18F]Flutemetamol, [11C]PIB, [11C]raclopride, [15O]H2O etc.. i.e. the used radionuclide in square brackets and 18, 11, etc. as superscripts. But without square brackets if e.g. 18F-labelled. The full chemical name has been added to [18F]FDG and [11C]CFT but not to the other tracers. Please check this inconsistency. Please use the IUPAC-name if possible; e.g. 2-Deoxy-2-[18F]fluoroglucose for [18F]FDG.

 - Reply: We thank the Reviewer. As suggested, the nomenclature of presented radiopharmaceuticals is now reported according to the guidelines presented in “Consensus nomenclature rules for radiopharmaceutical chemistry”. The full chemical name (IUPAC-name) is written for each radiotracer presented.

- Lines 72 and 73, 204 and 205, 393, 412. Please correct the name for 11C-labelled 2β-carbomethoxy-3β-(4-fluorophenyl)tropane. Instead of the letter b, the Greek letter beta should be used. In the text the name should be mentioned only once and thereafter to use the abbreviation [11C]CFT.

- Reply: We revised the manuscript as suggest by the Reviewer. The full chemical name is now used only once.

- The authors mention also a tau-tracer, S-[18F]THK-5117. This could perhaps be addressed separately. It may also be worth mentioning that this tracer has subsequently been shown to bind to MAO-B as well. In other words, the study should be repeated with some other, more specific tau tracer.

- Reply: We thank the Reviewer for the suggestion. The following sentence is now added at page 4: “However, this tau tracer binds also to monoamine oxidase-B (MAO-B) enzyme, which is known to be elevated in basal ganglia and could have led to off-target binding. This could constitute a limitation in the specificity of the study. Further studies with more specific Tau-tracer should be conducted.”

- Keywords: Please check if the keywords are the best ones.

- Reply: We thank the reviewer for the suggestion. We deleted the “glucose” keyword, inserting the keyword “biomarkers”.

- Line 138. Misspelled Rinne et al. (147)

- Reply: Thanks. We have corrected the typo.

- PiB has been written in two different ways. PiB is more common than Pib. Please check

- Reply: We have used “PiB” along the entire revised manuscript. 

- Line 243 and table 2, please check if there should be [15O]CO instead of [11C]CO. (Ref. 32 Brooks et al.)

- Reply: We have checked and we confirm that Brooks et al. used [11C]CO and not [15O]CO.

- Please check that all reference titles are written exactly like in the original paper, e.g. that the superscripts and square brackets are correct. Please, also use the method described in the IJMS to abbreviate the journal names.

- Reply: We have amended the reference section.

Reviewer 2 Report

In this manuscript, the authors reviewed the literature to assess the contribution of PET, PET/CT imaging with different radiotracers in the evaluation of patients with idiopathic normal pressure hydrocephalus.

It is rather a confused and not well organized manuscript and the studies reviewed in each functional aspect of PET imaging (beta amyloid, perfusion, glucose metabolism, dopaminergic function) do not clearly correlate with the conclusions of authors at the end of each aspect section. There is great heterogeneity in studies with respect to number of patients, PET evaluation techniques and timing of imaging. For example several perfusion and glucose metabolism PET studies were performed either before or after the surgical procedure; this could affect the prognostic value of PET imaging. It is not mentioned if all patients in the reviewed studies had cognitive impairment, as well as if the age of patients in these studies was taken into consideration (for example in beta amyloid imaging, the amyloid burden was compatible with AD or could be considered within normal limits in relation with the patients’ age?). Moreover, a combination of beta amyloid and glucose metabolism imaging could be more useful in the evaluation of concomitant AD in idiopathic normal pressure hydrocephalus. Additionally, not all patients in the reviewed studies had idiopathic normal pressure hydrocephalus. Biopsy specimens were not taken from the same brain region in all studies which could affect the evaluation of beta amyloid burden.

Language improvement is needed. In several parts of the manuscript it does not make sense (for example lines 161-162: …iNPH patients showed an increased cortical 11C-PiB retention and a lower with matter 11C-PiB retention in WM,…).

Several references are not written correctly. For example ref [17] – Wolk DA, co-authors are not mentioned in references section.

For all the above reasons, this manuscript does not contribute significantly to the field and I suggest that it is not suitable for publication in IJMS, in the existing form.

Author Response

- It is rather a confused and not well organized manuscript and the studies reviewed in each functional aspect of PET imaging (beta amyloid, perfusion, glucose metabolism, dopaminergic function) do not clearly correlate with the conclusions of authors at the end of each aspect section.

- Reply: We are sorry if the reviewer found our manuscript as confused and not well organized, as we did our best to summarize the multifaced possible role of PET imaging in NPH using several subheadings to group literature. We have simplified the manuscript deleting all the summary paragraphs and creating only a summary paragraph at the end of the revised manuscript.

- There is great heterogeneity in studies with respect to number of patients, PET evaluation techniques and timing of imaging. For example several perfusion and glucose metabolism PET studies were performed either before or after the surgical procedure; this could affect the prognostic value of PET imaging.

- Reply: According to the reviewer’ suggestion, we have added a “Limitations” section to discuss the suggested limitations, in particular the heterogenity among the studies. We recognize the limitation due to the heterogeneity of studies included in the review and extensively discuss it through the manuscript. Regarding prognostic value of PET, we clearly subdivided perfusion and metabolic studies according to their timing respect to shunting (please refer to subheadings 3.2.1/3.2.2/3.2.3 and 3.3.1/3.3.2). Of course the aim of these studies was different, as preoperative only studies aimed to detect the static picture of the disease and those possible markers of response to shunt, while pre-post-op studies aimed to increase knowledge on treatment induced changes and to detect those dynamic changes which could predict outcome at longer term.

- It is not mentioned if all patients in the reviewed studies had cognitive impairment, as well as if the age of patients in these studies was taken into consideration (for example in beta amyloid imaging, the amyloid burden was compatible with AD or could be considered within normal limits in relation with the patients’ age?).

- Reply: Not all the studies give explicit comments on the cognitive status of patients, or selected patients according to their cognitive status. However, in the great majority of papers some details on cognitive status measured to common neuropsychological tests as MMSE are furnished. A great heterogeneity in these scores exists among single patients even in the same study.

- A common limitation of the retrieved studies is the lack of detailed comparison of cognitive between different pathologies, namely iNPH and AD, and between patients with different imaging findings.

- Reply: We have discussed the aspect cited above in the limitations section.

- Moreover, a combination of beta amyloid and glucose metabolism imaging could be more useful in the evaluation of concomitant AD in idiopathic normal pressure hydrocephalus.

- Reply: We agree with this consideration. We have added the following sentence to the “future perspectives” section: “Moreover, studies with a combination of beta amyloid and glucose metabolism imaging could be useful in order to evaluate concomitant AD in idiopathic normal pressure hydrocephalus and to improve the possibility to assess the expected outcomes of surgical treatment.”

- Additionally, not all patients in the reviewed studies had idiopathic normal pressure hydrocephalus.

- Reply: The great majority of patients were affected by iNPH. However, we considered only iNPH patients when analyzing papers with mixed pathology. In some papers the generic term chronic hydrocephalus was used; in that case an accurate analysis of the manuscript to detect those patients which could be defined as iNPH was performed. We specified this aspect in the methods section of the revised manuscript. We have added specific sentences to comment different results of iNPH and secondary NPH where needed.

- Biopsy specimens were not taken from the same brain region in all studies which could affect the evaluation of beta amyloid burden.

- Reply: We have included this limitation in the appropriate section. We agree that the estimation of prevalence of beta amyloid deposit burden could be limited by heterogeneous brain tissue sampling. However, the aim of some of those papers was to validate beta amyloid tracers using biopsy samples as reference standard, while the aim of others was to assess whether presence/absence of these deposit was associated to outcome after shunting.

- Language improvement is needed. In several parts of the manuscript it does not make sense (for example lines 161-162: …iNPH patients showed an increased cortical 11C-PiB retention and a lower with matter 11C-PiB retention in WM,…).

- Reply: We have revised the manuscript taking into account the reviewer’s suggestion about language improvement.

- Several references are not written correctly. For example ref [17] – Wolk DA, co-authors are not mentioned in references section.

- Reply: We amended the reference section in the revised manuscript taking into account the reviewer’s comment.

Round 2

Reviewer 2 Report

The authors have improved significantly the manuscript which now warrants publication in IJMS.